# Parental vaccine hesitancy and acceptance of a COVID-19 vaccine: An internet-based survey in the US and five Asian countries

Grace Joachim[1☯], Shu-Fang Shih[2☯], Awnish Singh[3], Yogambigai Rajamoorthy[4], Harapan Harapan[5,6,7,8], Hao-Yuan Chang[9,10], Yihan Lu[11], Abram L. Wagner[12]*

1 Department of Epidemiology and Biostatistics, College of Human Medicine, Michigan State University, East Lansing, Michigan, United States of America, 2 Department of Health Administration, College of Health Professions, Virginia Commonwealth University, Richmond, Virginia, United States of America, 3 National Technical Advisory Group on Immunisation Secretariat, National Institute of Health and Family Welfare, New Delhi, India, 4 Department of Economics, Universiti Tunku Abdul Rahman, Selangor, Malaysia, 5 Medical Research Unit, School of Medicine, Universitas Syiah Kuala, Banda Aceh, Indonesia, 6 Department of Microbiology, School of Medicine, Universitas Syiah Kuala, Banda Aceh, Indonesia, 7 Tropical Disease Centre, Universitas Syiah Kuala, Banda Aceh, Indonesia, 8 Tsunami & Disaster Mitigation Research Center (TDMRC), Universitas Syiah Kuala, Banda Aceh, Indonesia, 9 School of Nursing, College of Medicine, National Taiwan University, Taipei, Taiwan, 10 Department of Nursing, National Taiwan University Hospital, Taipei, Taiwan, 11 Key Laboratory of Public Health Safety (Ministry of Education), Department of Epidemiology, Fudan University School of Public Health, Shanghai, China, 12 Department of Epidemiology, School of Public Health, University of Michigan, Ann Arbor, Michigan, United States of America

☯ These authors contributed equally to this work.
* awag@umich.edu

**Data Availability Statement:** The data collected from these surveys have been made publicly

## Abstract

COVID-19 vaccination rates for children globally are relatively low. This study aimed to investigate parental vaccine hesitancy and parents' acceptance of a COVID-19 for their children for their children in the United States, China, Taiwan, India, Indonesia, and Malaysia. We analyzed data from an opt-in, internet-based cross-sectional study (n = 23,940). Parents were asked about their acceptance of a COVID-19 vaccine for their children, and if they would accept the vaccine with different risk and effectiveness profiles for themselves. Poisson regression was used to generate prevalence ratios (PR) of the relationship between vaccine acceptance for a child and vaccine profile, by country and waves and overall. Between August 2020 and June 2021, COVID-19 vaccine acceptance for children decreased in the United States (89% to 72%) and Taiwan (79% to 71%), increased in India (91% to 96%) and Malaysia (81% to 91%), and was stable in Indonesia (86%) and China (at 87%-90%). Vaccine risk and effectiveness profiles did not consistently affect parent's acceptance of a COVID-19 vaccine for their children. Instead, being not hesitant was a large driver of vaccine acceptance (PR: 1.24, 95% CI: 1.14, 1.36). Adolescent COVID-19 vaccination have already been established in many high and middle-income countries, but our study suggests that there is a movement of vaccine hesitancy which could impede the success of future pediatric and adolescent COVID-19 vaccination programs.

available and can be accessed at https://doi.org/10.3886/E130422V2.

**Funding:** This project was supported by an award from the National Science Foundation, Division of Social and Economic Sciences (#2027836 to AW, https://www.nsf.gov/awardsearch/showAward?AWD_ID=2027836). This project funded all co-authors on this manuscript. The content is solely the responsibility of the authors and does not necessarily represent the official views of the National Science Foundation. The funders had no role in study design, data collection and analysis, decision to publish, or preparation of the manuscript.

**Competing interests:** The authors have declared that no competing interests exist.

## Introduction

In 2020, the emergence of severe acute respiratory syndrome coronavirus 2 (SARS-CoV-2) led to a global health crisis, with many countries declaring a state of emergency. The first case of this new virus was reported in the United States (US) in February 2020, and on March 11, 2020, the World Health Organization declared a pandemic [1]. As of December 13, 2022, there have been almost 1,300 million confirmed cases of coronavirus disease 2019 (COVID-19) worldwide [2].

According to recent statistics released by the American Academy of Pediatrics, children represented about 18.2% of all COVID-19 cases in the US, with an overall rate of 20,096 cases per 100,000 children in the population [3, 4]. Children ranged from 1.6%-4.3% of total accumulated hospitalizations, and 0.1%-2.0% of all child COVID-19 cases resulted in hospitalization based on the data reported by 24 states and New York City from May 2021 to October 21, 2021 [3].

Vaccines can be an important way to protect against serious illness from COVID-19, but the vaccine became available for children later than for adults. Parents may have differing preferences in vaccinating themselves compared to their children, particularly when it comes to vaccine safety [5].

Surveys from the US have examined parents' attitudes and acceptance of a COVID-19 vaccine for their children. According to surveys conducted by the Kaiser Family Foundation (KFF) prior to rollout of the vaccine in children, only approximately 26% to 34% of parents have indicated that they will definitely get their 5–11 years old children vaccinated against COVID-19. In addition, 31% to 35% of parents have stated that they would have their children vaccinated only if it is mandatory or that they definitely do not want their children to receive the vaccine [6]. Another survey conducted by Gallup between May and October 2021 showed that, on average, 45% of parents would not be willing to have their children under 12 receive the COVID-19 vaccine–a higher percentage than what has been found in KFF surveys [7].

Actual vaccination coverage in children has varied across countries. In the US, the vaccination rate for children aged 12–17 years was 71.4%, 39.4% for children aged 5–11 years, 9.7% for children aged 2–4 years, and 6.8% for children under 2 years old [8]. This rate is much lower compared with the vaccination rate of at least first dose among adults aged 25 to 49 (84.8%), the age range of most parents of young children or adolescents [8].

In comparison, the vaccination rate among children aged 12–17 outside of the United States is higher in some countries than that in the United States. China began administering COVID-19 vaccines to children aged 12–17 years in July 2021 and to children aged 3–11 years in October 2021. According to the most recent publicly available data, 91% of students aged 12–17 in September 2021 were fully vaccinated in China [9]. In Taiwan, since September 23, 2021, the Ministry of Health and Welfare and the Ministry of Education has been vaccinating high school students [10]. As of November 14, 2022, 95.3% of children 12–17 in Taiwan had received at least one dose [11]. In Malaysia, the vaccination rate for the first dose among students aged 12–17 is 94.6%, and 50% for students aged 5 to 11 years old as of October 6, 2022 [12]. In Indonesia, the vaccination for children (aged 12–17 years old) started on July 1, 2021, and as of May 18, 2023, 83.60% of the targeted population have been vaccinated [13].

Concerns surrounding vaccine safety have been present since the first smallpox immunization campaigns [14]. For the current COVID-19 vaccine, various studies have attempted to examine the factors contributing to parents' hesitancy towards vaccination for their children. For example, one study examined the relationship between demographic characteristics, like parents' gender, race/ethnicity, education, and political affiliation, and propensity to vaccinate a child against COVID-19 at an early phase in the vaccine rollout [15]. A more global question

is what are the patterns of general vaccine hesitancy and acceptance of a COVID-19 vaccine across countries, and do these patterns of vaccine acceptance vary by socioeconomic characteristics. Therefore, the objective of this study is to investigate parental vaccine hesitancy and parents' acceptance of a COVID-19 for their children in selected countries.

## Materials and methods

### Study population

This study used an opt-in, internet-based sample that was recruited through social media and online advertisements by a survey research firm. Cross-sectional, online surveys were conducted in six countries, including the US, China, Indonesia, India, Malaysia, and Taiwan in August 2020, November 2020, March 2021, and June 2021. An additional survey was conducted in China in March 2020 and in the US in June 2020, October 2020, February 2021, and April 2021. The eligibility criteria included being an adult residing in the country where the data were collected. For each wave, we attempted to obtain a sample size of 800, in order to estimate an outcome proportion of 50% (a statistically conservative estimate of the population vaccinated), based on a margin of error was 4% and with an alpha of 0.05 and a power of 80%. The data collected from these surveys have been made publicly available and can be accessed at https://doi.org/10.3886/E130422V2. Research staff did not ever have access to personally identifiable information from study participants. Information about our approach to inclusivity in global research is included in the **S1 Text**.

### Measurement

This study includes different vaccine-related measures, including hypothetical acceptance of a vaccine for oneself vs a child, actual vaccination behaviors for oneself, and hesitancy towards adult vaccines in general–with the perspective of vaccine hesitancy as a psychological state of indecision [16].

Before the vaccine was available, we asked about hypothetical acceptance of a COVID-19 vaccine with a given *vaccine profile*. Individuals were randomized to receive one of four sets of profiles, which differed by safety and effectiveness (50% effective with a 20% risk of fever; 50% effective with a 5% risk of fever; 95% effective with a 20% risk of fever; or 95% effective with a 5% risk of fever). These estimates were chosen for the questionnaire in 2020 and reflect some of the range of vaccine effectiveness seen in other vaccines, with a lower bound from the influenza vaccine [17] and the upper bound from the measles vaccine [18].

Starting spring 2021, participants were instead asked whether they had received a coronavirus vaccine, planned to receive a vaccine, or had already been vaccinated.

During the survey, participants were asked if they had children under the age of 18. If they responded in the affirmative, they were then given a vaccine profile (the same as for the adult vaccine, which varied effectiveness and safety) and asked a question about their acceptance of a coronavirus vaccine for their children: "Would you accept a coronavirus vaccine for your child?"

Using these two variables, we created a variable to examine discordant vaccine acceptance between the parent and child (wanting a vaccine for self and child, wanting a vaccine for self but not for child, wanting a vaccine for child but not for self, or not wanting a vaccine for self or child).

We also measured general, adult vaccine hesitancy using the validated 10-item adult Vaccine Hesitancy Scale (aVHS) [19]. The aVHS had a 5-point Likert scale as answer choices, ranging from least hesitant (1) to most hesitant (5). Based on a published standard, we dichotomized this variable into vaccine hesitant and non-vaccine hesitant categories [19].

## Statistical analysis

We used a Poisson regression model with robust variance estimators to output prevalence ratios (PRs) for vaccine acceptance for a child in each country and wave in this study, using the vaccine profile as the independent variable.

Subsequently, in a model that included all countries, we also used Poisson regression to estimate PRs and 95% confidence intervals (CIs) for the association between vaccine acceptance for a child and the vaccine profile, vaccine hesitancy, education level, and the month of the survey.

In an unadjusted analysis, we estimated the frequency of discordant vaccine views between self and child by vaccine hesitancy and education level, separately for each country in the June 2021 wave. We assessed significance using a Rao-Scott chi-square test or Fisher's exact test. We used only one wave out of concern that acceptance and hesitancy regarding COVID-19 vaccines could vary over time.

Finally, we include a measure of population attributable fraction, relating general vaccine hesitancy and non-vaccination of children [20]. Briefly, we used log binomial models and the frequency of non-vaccination of children with vaccine-hesitant parents to estimate the fraction of non-vaccination of children that was related to parental vaccine hesitancy.

Individuals with missing data were excluded from analysis. The data were weighted to be representative of national populations in terms of age, gender, and race. We conducted our analyses using SAS Version 9.4 (Cary, North Carolina).

## Ethical approval

The protocol was reviewed and approved by ethical review committees in each of the six countries, including the University of Michigan Health Sciences and Behavioral Sciences Institutional Review Board (#HUM00180096), the Fudan University School of Public Health ethical review committee (#IRB00002408), the National Taiwan University Hospital Research Ethics Committee (#202007102RINB), the Universiti Tunku Abdul Rahman (#U/SERC/107/2020), the Komite Etik Penelitian Kesehatan at Universitas Syiah Kuala (#041/EA/FK-RSUDZA/2020), and the Sigma-IRB in New Delhi, India (#10003/IRB/20-21). Prior to participating in the study, participants were provided with an informed consent to read and review. They were asked to click "I agree to participate in the study" button prior to any data collection occurring.

## Results

The sample size for this analysis was 23,940 participants across all waves and countries. Each wave of data collection included over 630 participants. More information on the sample size, the number of participants who agreed to participate in this study, and the number of participants who completed the study can be found online (https://doi.org/10.6084/m9.figshare.14792058.v3). For the US participants, we used eight waves of surveys, while for China, we used five waves of surveys, and for the other four countries, we used four waves of surveys.

Fig 1 shows the distribution of vaccine acceptance for children by country and wave. In the US, parent's acceptance of a COVID-19 vaccine for their children ranged from 90% in June 2020 to 67% in March 2021. This indicated an overall decrease in acceptance across all waves. The trend of declining vaccine acceptance over time was also observed in Taiwan, where acceptance ranged from 64% in March 2021 to 79% in August 2020. Conversely, in China and Indonesia, acceptance remained relatively stable around 90% and 86%, respectively, across all waves. India and Malaysia demonstrated an overall increase in vaccine acceptance over time.

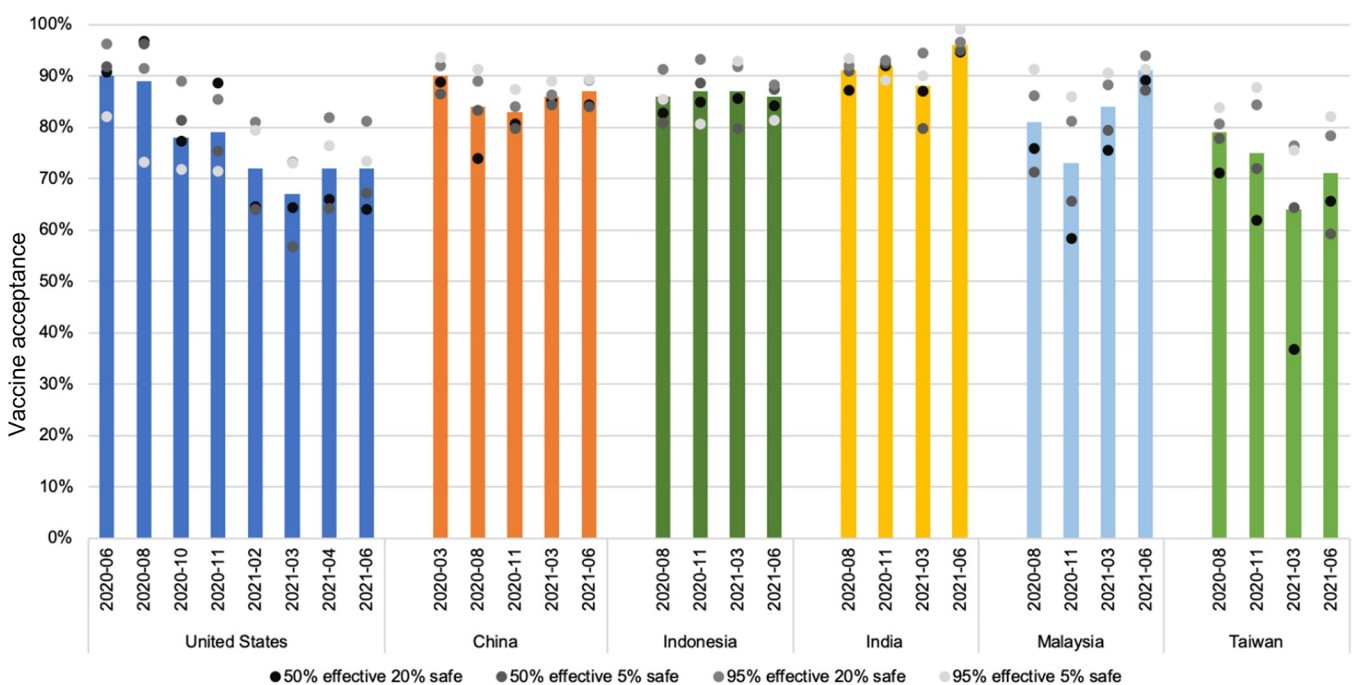

**Fig 1. Percentage of COVID-19 vaccine acceptance for child by wave and country.**

The population attributable fraction of non-vaccinated children with vaccine hesitant parents varied across time are showed in Fig 2. We see noticeable increases in the fraction for the US, Taiwan, and Indonesia over time, which suggests that in these locations, vaccine hesitancy became a larger driving force for parental acceptance for pediatric vaccines at later times. In the US, the population attributable fraction ranged from 3.7% in August 2020 to 26.3% in February 2021. In contrast, the fraction's range was smaller in China, from 1.5% in March 2021 to 8.4% in November 2020. The population attributable fraction peaked in Indonesia (23.0%), India (8.7%), Malaysia (13.2%), and Taiwan (36.1%) in March 2021, which was similar to the peak observed in the US in February 2021 (26.3%).

According to Table 1, we observed variations in vaccine hesitancy across countries and waves. Specifically, in the US, vaccine hesitancy ranged from 40% in June 2021 to 60% in June 2020. Similarly, in Taiwan, hesitancy ranged from 46% in June 2021 to 59% in March 2021. In other countries however, demonstrated lower levels of vaccine hesitancy. For instance, China exhibited vaccine hesitancy levels ranging from 22% in March 2020 and June 2021 to 30% in November 2020.

Our study found that the vaccine profile did not consistently influence vaccine acceptance (Table 2). In 14 out of the 29 waves of data collection, there were no significant effects. However, Taiwan consistently showed strong preferences for a safer and more effective vaccine, with a 22% lower rate of preferring the 50% effective and 20% safe vaccine In June 2021 compared to the 95% effective and 5% safe vaccine (p = 0.006). In the United States, there were stronger preferences over time for a safer and more effective vaccine, with a 12% lower rate of preferring the 50% effective and 20% safe vaccine in June 2021 compared to the 95% effective and 5% safe vaccine (p = 0.037).

Table 3 shows that parents from all countries generally preferred a safer and more effective vaccine for their children. The prevalence of accepting a vaccine for a child was 1.24 times higher among parents who were not hesitant about vaccines compared to those who were

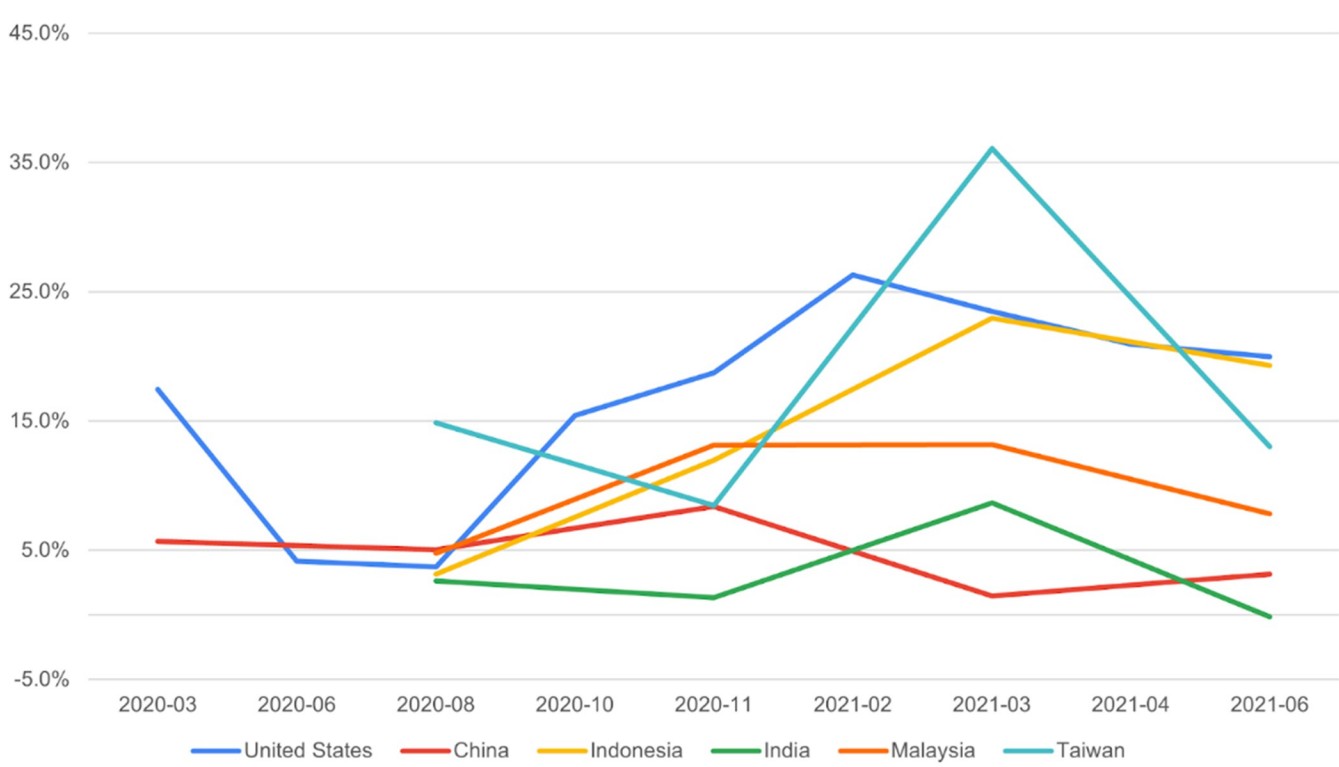

**Fig 2. Population attributable fraction of non-vaccination of children among vaccine-hesitant parents across waves and countries.**

hesitant (95% CI: 1.14, 1.36). Furthermore, there was no significant difference in vaccine acceptance for children between those with and without college education (PR: 1.02, 95% CI: 0.93, 1.12).

The degree of vaccine preference discordance in children and adults is shown in Table 4. Notably, a relatively low proportion in the US and Taiwan wanted a vaccine for themselves and for their children (65% in the US and 62% in Taiwan). In contrast, in other countries, over 80%, of participants expressed willingness for vaccination for themselves and their children. Across all countries, individuals classified as vaccine hesitant were found to be less likely to vaccinate themselves across countries. However, in some situations, they expressed different views regarding the vaccination of themselves versus their children. For instance, in the US and Malaysia, those who were hesitant were more likely to want a vaccine for their children, but not for themselves. Finally, our study revealed a significant difference in vaccination preferences among participants and their children in the US and Taiwan based on college education.

## Discussion

In this large, repeated cross-sectional study, encompassing six different countries and multiple waves of data, we identified discernible variations in COVID-19 vaccine acceptance for children by location and time. Specifically, we found that vaccine acceptance for children decreased in the US and Taiwan, increased in Malaysia and India, and stayed stable in China and Indonesia. Across countries, general vaccine hesitancy strongly correlated with patterns of parental vaccine decision-making, whereas education, our proxy for socioeconomic status, only significantly correlated with vaccine decision-making in the US and Taiwan. Overall, our

**Table 1. Descriptive statistics of vaccination outcomes by country and survey wave (N = 23,940).**

| Country/Year & Month | Vaccine hesitant | Wants vaccine for self | Wants vaccine for child | Actual vaccination status for self | | |
|---|---|---|---|---|---|---|
| | | | | No plan to get vaccinated | Plans to get vaccinated | Already vaccinated |
| **US** | | | | | | |
| 2020–06 | 55% | 84% | 90% | - | - | - |
| 2020–08 | 41% | 81% | 89% | - | - | - |
| 2020–10 | 50% | 61% | 78% | - | - | - |
| 2020–11 | 54% | 65% | 79% | - | - | - |
| 2021–02 | 50% | - | 72% | 37% | 41% | 23% |
| 2021–03 | 50% | - | 67% | 34% | 27% | 39% |
| 2021–04 | 43% | - | 72% | 22% | 13% | 65% |
| 2021–06 | 40% | - | 72% | 23% | 10% | 67% |
| **China** | | | | | | |
| 2020–03 | 22% | 96% | 90% | - | - | - |
| 2020–08 | 24% | 94% | 84% | - | - | - |
| 2020–11 | 30% | 91% | 83% | - | - | - |
| 2021–03 | 28% | - | 86% | 4% | 30% | 66% |
| 2021–06 | 22% | - | 87% | 2% | 5% | 93% |
| **Indonesia** | | | | | | |
| 2020–08 | 38% | 91% | 86% | - | - | - |
| 2020–11 | 44% | 87% | 87% | - | - | - |
| 2021–03 | 32% | - | 87% | 14% | 49% | 37% |
| 2021–06 | 37% | - | 86% | 12% | 24% | 65% |
| **India** | | | | | | |
| 2020–08 | 29% | 95% | 91% | - | - | - |
| 2020–11 | 32% | 94% | 92% | - | - | - |
| 2021–03 | 38% | - | 88% | 8% | 35% | 57% |
| 2021–06 | 28% | - | 96% | 2% | 10% | 88% |
| **Malaysia** | | | | | | |
| 2020–08 | 33% | 86% | 81% | - | - | - |
| 2020–11 | 42% | 81% | 73% | - | - | - |
| 2021–03 | 36% | - | 84% | 18% | 60% | 23% |
| 2021–06 | 33% | - | 91% | 10% | 50% | 40% |
| **Taiwan** | | | | | | |
| 2020–08 | 50% | 84% | 79% | - | - | - |
| 2020–11 | 49% | 79% | 75% | - | - | - |
| 2021–03 | 59% | - | 64% | 47% | 43% | 9% |
| 2021–06 | 46% | - | 71% | 20% | 55% | 24% |

results point to the need to consider vaccine hesitancy as a global phenomenon with local manifestations and consequences.

By examining country-specific vaccination patterns over time, we could tie certain attitudes to policies. For example, in Malaysia, the initiation of the adult COVID-19 vaccination program in February 2021 led to an increase in parents' confidence in vaccinating their children, rising from 81% in August 2020 to 84% in March 2021 [12]. Moreover, a study conducted after the duration of this study also proved that parents COVID-19 vaccination history strongly influenced the parent's discussion on vaccinating their children [21]. Therefore, this could be an example of parental familiarity with the COVID-19 vaccine positively impacting attitudes

**Table 2. Poisson regression results: Parent's COVID-19 vaccine acceptance for their child based on randomized vaccine effectiveness and risk profiles.**

| Country/Year & Month | N | Vaccine Profiles | | | P-value |
|---|---|---|---|---|---|
| | | odds ratio (reference: 95% effective with a 5% risk of fever) | | | |
| | | 50% effective with a 20% risk of fever | 50% effective with a 5% risk of fever | 95% effective with a 20% risk of fever | |
| **US** | | | | | |
| 2020–06 | 657 | 1.06 | 1.13 | 1.14 | 0.267 |
| 2020–08 | 783 | 1.25* | 1.26* | 1.17 | 0.03 |
| 2020–10 | 937 | 1.14 | 1.24* | 1.24* | 0.1 |
| 2020–11 | 986 | 1.24* | 1.1 | 1.22* | 0.061 |
| 2021–02 | 877 | 0.82 | 0.79* | 1.02 | 0.026 |
| 2021–03 | 917 | 0.88 | 0.77* | 0.98 | 0.114 |
| 2021–04 | 917 | 0.86 | 0.81 | 1.06 | 0.045 |
| 2021–06 | 954 | 0.88 | 0.93 | 1.12 | 0.037 |
| **China** | | | | | |
| 2020–03 | 1070 | 0.95 | 0.92* | 0.97 | 0.175 |
| 2020–08 | 788 | 0.83** | 0.90* | 0.97 | 0.002 |
| 2020–11 | 939 | 0.92 | 0.93 | 0.99 | 0.299 |
| 2021–03 | 721 | 0.96 | 0.93 | 0.95 | 0.512 |
| 2021–06 | 971 | 0.94 | 0.94 | 1.02 | 0.162 |
| **Indonesia** | | | | | |
| 2020–08 | 727 | 0.94 | 0.99 | 1.07 | 0.192 |
| 2020–11 | 800 | 1.03 | 1.08 | 1.14* | 0.032 |
| 2021–03 | 789 | 0.94 | 0.90* | 1.01 | 0.042 |
| 2021–06 | 783 | 0.98 | 0.97 | 1.01 | 0.856 |
| **India** | | | | | |
| 2020–08 | 805 | 0.95 | 0.96 | 0.98 | 0.548 |
| 2020–11 | 957 | 1.03 | 1.01 | 1.03 | 0.786 |
| 2021–03 | 926 | 0.98 | 0.90* | 1.04 | 0.007 |
| 2021–06 | 894 | 0.95* | 0.95* | 0.98 | 0.039 |
| **Malaysia** | | | | | |
| 2020–08 | 759 | 0.84* | 0.83* | 0.95 | 0.015 |
| 2020–11 | 738 | 0.74** | 0.84* | 0.93 | 0.001 |
| 2021–03 | 749 | 0.82** | 0.85* | 1.01 | 0 |
| 2021–06 | 779 | 0.96 | 0.96 | 1.02 | 0.4 |
| **Taiwan** | | | | | |
| 2020–08 | 645 | 0.82 | 0.87 | 0.9 | 0.287 |
| 2020–11 | 633 | 0.67** | 0.82* | 0.95 | 0.001 |
| 2021–03 | 679 | 0.54*** | 0.84 | 0.98 | 0 |
| 2021–06 | 760 | 0.78* | 0.73* | 0.95 | 0.006 |

Note

*p<0.05

**: p<0.01

***: p<0.001

towards the pediatric vaccine. There is already a growing body of work on how experiences of a vaccine-preventable disease can positively, or negatively, impact acceptance of a vaccine [22].

Our measure of a population attributable fraction for vaccination quantifies how much vaccine hesitancy impacted pediatric vaccine acceptance. Studies in the US, for example, have

**Table 3. Multivariable model results: Parent's COVID-19 vaccine acceptance for their child across countries and waves (N = 19,482).**

| Research Variables | Vaccine Acceptance Prevalence Ratio (95% CI) |
| --- | --- |
| **Vaccine Profile (reference: 95% effective, 5% risk of fever)** | |
| 50% effective, 20% risk of fever | 0.91 (0.81, 1.01) |
| 50% effective, 5% risk of fever | 0.92 (0.82, 1.02) |
| 95% effective, 20% risk of fever | 1.01 (0.90, 1.12) |
| **Vaccine hesitancy (reference: Hesitant)** | |
| Not hesitant | 1.24 (1.14, 1.36) |
| **Education (reference: Some college)** | |
| High school or less | 1.02 (0.93, 1.12) |
| **Country (reference: US)** | |
| China | 1.20 (1.03, 1.40) |
| Indonesia | 1.23 (1.06, 1.42) |
| India | 1.31 (1.13, 1.51) |
| Malaysia | 1.23 (1.06, 1.43) |
| Taiwan | 1.01 (0.85, 1.21) |
| **Wave of Data Collection (reference: 2021–06)** | |
| 2021–03 | 0.97 (0.89, 1.04) |

examined regional differences in the impact of parental vaccine hesitancy on childhood vaccination through this method [23]. In the US, Taiwan, and Indonesia, we found more notable increases in the fraction, indicating that vaccine hesitancy was a larger driver for parental acceptance for pediatric vaccines at later time points. Other drivers could include things like– lack of perception of need for the vaccine or the belief that the disease is not severe in children [24]. We note also that other studies have showed that parental perceptions of vaccine safety and effectiveness in children have a large impact on acceptance. A study conducted in Taiwan from July to September 2021 also found a relatively large proportion (approximately 64%) expressing reservations about vaccinating their children [25]. Notably, this study identified perceived vaccine safety and the preventative efficacy against COVID-19 as significant factors contributing to parental vaccine hesitancy [25].

We used education to study the relationship between socioeconomic status and vaccination attitudes because education was able to be measured relatively consistently across countries. Moreover, other studies investigating COVID-19 vaccine acceptance have reported education as a significant determinant [26, 27]. We note that education seemingly is country-specific in terms of its correlation with vaccination patterns. In our study it had the largest impact in the US and Taiwan. These also are the wealthiest countries in terms of average income in our study, and it is possible that there are different levels of interaction between education, income, wealth, and other socioeconomic variables that we were unable to evaluate in our study.

Our study examined the concordance between parental vaccination acceptance for themselves and their children. This was also the subject of another study, in Greece [28]. One assumption is that parents are more risk adverse for pediatric vaccines [5]. We found, in the US, China, and Taiwan that individuals were much more likely to want vaccines for themselves than for their children, if there was any discordance in vaccine acceptance between self and the child. This potentially could be tied to more concerns about vaccine safety or perceptions of the disease being less severe in children than adults.

Cultural differences could have played a substantial role in shaping the observed differences in vaccination between countries. This could relate in part due to individualistic vs collectivist

**Table 4. Frequency of discordant coronavirus vaccination views between parents and children by vaccine hesitancy and college education, June 2021 (N = 5,141).**

| Country and subpopulation | Want vaccines for self and child | Want vaccines for self, not child | Want vaccine for child, not self | Do not want vaccine for either | P-value[a] |
|---|---|---|---|---|---|
| **US** | | | | | |
| Overall | **65%** | **12%** | **7%** | **17%** | |
| General vaccine hesitancy | | | | | <0.001 |
| Vaccine hesitant | 44% | 7% | 12% | 37% | |
| Not vaccine hesitant | 79% | 15% | 4% | 3% | |
| Education | | | | | <0.001 |
| College education | 70% | 14% | 5% | 11% | |
| No college education | 49% | 7% | 12% | 33% | |
| **China** | | | | | |
| Overall | **86%** | **12%** | **1%** | **1%** | |
| General vaccine hesitancy | | | | | 0.017 |
| Vaccine hesitant | 80% | 15% | 1% | 5% | |
| Not vaccine hesitant | 87% | 11% | 1% | 1% | |
| Education | | | | | 0.143 |
| College education | 85% | 13% | 1% | 2% | |
| No college education | 93% | 6% | 1% | 0% | |
| **Indonesia** | | | | | |
| Overall | **81%** | **7%** | **4%** | **7%** | |
| General vaccine hesitancy | | | | | <0.001 |
| Vaccine hesitant | 61% | 12% | 10% | 18% | |
| Not vaccine hesitant | 93% | 4% | 1% | 1% | |
| Education | | | | | 0.227 |
| College education | 83% | 7% | 3% | 7% | |
| No college education | 76% | 7% | 8% | 10% | |
| **India** | | | | | |
| Overall | **95%** | **3%** | **1%** | **1%** | |
| General vaccine hesitancy | | | | | 0.007 |
| Vaccine hesitant | 92% | 3% | 3% | 1% | |
| Not vaccine hesitant | 96% | 3% | 1% | 1% | |
| Education | | | | | 0.323 |
| College education | 94% | 4% | 1% | 1% | |
| No college education | 97% | 2% | 1% | 0% | |
| **Malaysia** | | | | | |
| Overall | **84%** | **6%** | **6%** | **3%** | |
| General vaccine hesitancy | | | | | <0.001 |
| Vaccine hesitant | 68% | 9% | 15% | 9% | |
| Not vaccine hesitant | 92% | 5% | 3% | 1% | |
| Education | | | | | 0.849 |
| College education | 85% | 6% | 7% | 3% | |
| No college education | 83% | 7% | 7% | 4% | |
| **Taiwan** | | | | | |
| Overall | **62%** | **19%** | **9%** | **10%** | |

(*Continued*)

**Table 4.** (Continued)

| Country and subpopulation | Want vaccines for self and child | Want vaccines for self, not child | Want vaccine for child, not self | Do not want vaccine for either | P-value[a] |
|---|---|---|---|---|---|
| General vaccine hesitancy | | | | | <0.001 |
| Vaccine hesitant | 46% | 18% | 18% | 19% | |
| Not vaccine hesitant | 75% | 21% | 2% | 2% | |
| Education | | | | | 0.029 |
| College education | 64% | 19% | 7% | 10% | |
| No college education | 48% | 18% | 23% | 10% | |

[a] from Rao-Scott chi-square test, except with cell counts <5, which used Fisher's exact test.

orientations of the culture. For example, in the US, where higher levels of vaccine hesitancy were identified, there is a more prevalent individualistic culture. Conversely, in China, where vaccine hesitancy was lower, a more collectivist culture framework prevails [29, 30]. Previous research has found that individualistic attitudes can contribute to vaccine hesitancy, whereas collectivist orientations tend to mitigate such hesitancy [31]. How this paradigm can be applied to each country, is uncertain, given large diversities of cultures within country. Substantial economic development in places like China, Malaysia, Indonesia, and India also may suggest upcoming changes in the individualistic-collectivist orientation.

Religion is another dimension of culture that could explain some across-country differences [32]. Historically, there has been lower pediatric vaccination coverage among Muslims in some areas [33]. In our study, Malaysia and Indonesia have Muslim-majority populations, and parents' acceptance of a vaccination is strongly influenced by the halal status of the vaccine [34]. Overall, the cultural, economic, and political differences across countries could influence individual attitudes and behaviors, and understanding these nuances is crucial in addressing vaccine hesitancy and promoting vaccine acceptance globally.

## Implications of research

Across the literature, health care providers remain an important and trusted source of information about vaccines [35], but this assumes health care providers have the time and training to discuss vaccine concerns with parents. For example, in an early study about vaccine decision-making, parents stated that they did not have enough time to talk with their doctors about their concerns with the measles, mumps, and rubella (MMR) vaccine [36]. Another qualitative study highlighted that the quality of the parent-provider relationship was especially key for moving forward vaccine decision-making on childhood vaccines [37]. Our study did not directly evaluate parent-provider interactions, but we do note there is large overlap in overall (adult) vaccine hesitancy and acceptance of a vaccine for a child. And in several countries, this relationship has grown stronger over time. Therefore, vaccine hesitancy should not just be understood as a circumstance of high-income countries, but one with potential impacts globally. This may necessitate further funding for pediatricians to have more time to have discussions with parents and for them to have training at effective communication strategies aimed at mitigating vaccine misconceptions. Building and maintaining trust between parents and healthcare providers is key to fostering vaccine confidence and acceptance.

Other community members could also be important sources of information and dialog about vaccines. Because religion also plays a significant role in shaping vaccine hesitancy among adults worldwide [38], religious leaders could be important vaccine messengers.

Other research has also focused on how vaccine hesitancy and vaccine misinformation is spread. The dissemination of misinformation through media channels has played a pivotal role in fueling vaccine hesitancy throughout the pandemic [25]. A US study conducted in October 2021 found that nearly 8 in 10 people either believe or expressed uncertainty about common myths surrounding COVID-19 or the vaccine, with unvaccinated adults exhibiting lower levels of trust in news sources for obtaining COVID-19 related information compared to their vaccinated counterparts [39]. News could influence vaccination through reporting of cases. Seeing a severe case of COVID-19 in the news has been associated with intent to vaccinate and actual vaccination status, and this association could be mediated by increased perceptions of susceptibility to illness [40].

How governments communicated to citizens could have also influenced the country specific trends we found. For example, in the US, the response to vaccination during the pandemic was often perceived as lacking coordination and clarity, with vaccination becoming intertwined with political ideologies for many individuals [41]. On the other hand, studies conducted in countries like China have found that government communication efforts were positively associated with vaccination intent [42].

There are systematic ways that governments can discover and respond to vaccine-related hesitancies and barriers. The Tailoring Immunization Programmes (TIP), developed by the World Health Organization, has demonstrated success in understanding the specific barriers to vaccination faced by different populations. By identifying and addressing these barriers, TIP aims to design vaccination programs that effectively meet the needs of diverse groups [43].

## Strengths and limitations

This study used data from an opt-in, internet-based sample, which may introduce bias and limit generalizability to the broader population. However, this sampling approach allowed for efficient and timely data collection, particularly, given the circumstances imposed by the pandemic. It is important to acknowledge that participants were required to have internet access to complete the survey, which may introduce a potential source of bias in the sample. In addition, the reliance on self-reported data may be subject to social desirability bias, potentially affecting the validity of responses. Our study also looks at vaccination on a national level, but there could be substantial differences subnationally, including in the relationship between socioeconomic status and vaccination [44]. Nevertheless, this study employed consistent survey methods across six countries and multiple waves, enabling meaningful comparisons of results both within and between countries over time. This approach provides valuable insights into cross-country variations and trends related to vaccine attitudes and behaviors, contributing to our understanding of the broader landscape of vaccine hesitancy.

## Conclusions

Early in the COVID-19 pandemic, it was already established that in many countries, a substantial proportion of the adult population would refuse a vaccine for themselves. Adolescent COVID-19 vaccination have already been established in many high and middle-income countries, but our study suggests that there is a movement of vaccine hesitancy which could impede the success of future pediatric and adolescent COVID-19 vaccination programs.

## Supporting information

**S1 Checklist. STROBE statement—checklist of items that should be included in reports of observational studies.**
(DOCX)

**S1 Text. Inclusivity in global research questionnaire.**
(DOCX)

## Acknowledgments

We would like to extend our gratitude to Mengdi Ji and Kaitlyn Akel for their valuable assistance in preparing the datasets.

## Author Contributions

**Conceptualization:** Shu-Fang Shih, Abram L. Wagner.

**Data curation:** Grace Joachim, Abram L. Wagner.

**Formal analysis:** Grace Joachim.

**Funding acquisition:** Shu-Fang Shih, Abram L. Wagner.

**Investigation:** Shu-Fang Shih, Abram L. Wagner.

**Methodology:** Shu-Fang Shih, Abram L. Wagner.

**Project administration:** Shu-Fang Shih, Awnish Singh, Yogambigai Rajamoorthy, Harapan Harapan, Hao-Yuan Chang, Yihan Lu, Abram L. Wagner.

**Writing – original draft:** Grace Joachim, Shu-Fang Shih.

**Writing – review & editing:** Grace Joachim, Shu-Fang Shih, Awnish Singh, Yogambigai Rajamoorthy, Harapan Harapan, Hao-Yuan Chang, Yihan Lu, Abram L. Wagner.

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
