## [Decision Letter · Decision Letter 0]

22 Aug 2023

PGPH-D-23-01104

Parents’ Vaccine Hesitancy and Behavior Affects Decision Making for Their Children's COVID-19 Vaccination in the US and Five Asian Countries

Dear Dr. Wagner,

Thank you for submitting your manuscript to PLOS Global Public Health. After careful consideration, we feel that it has merit but does not fully meet PLOS Global Public Health’s publication criteria as it currently stands. Therefore, we invite you to submit a revised version of the manuscript that addresses the points raised during the review process.

We look forward to receiving your revised manuscript.

Kind regards,

Julia Robinson

Executive Editor

Journal Requirements:

2. Please provide separate figure files in .tif or .eps format.

Additional Editor Comments (if provided):

Reviewers' comments:

Reviewer's Responses to Questions

**Comments to the Author**

1. Does this manuscript meet PLOS Global Public Health’s publication criteria? Is the manuscript technically sound, and do the data support the conclusions? The manuscript must describe methodologically and ethically rigorous research with conclusions that are appropriately drawn based on the data presented.

Reviewer #1: No

2. Has the statistical analysis been performed appropriately and rigorously?

Reviewer #1: Yes

3. Have the authors made all data underlying the findings in their manuscript fully available (please refer to the Data Availability Statement at the start of the manuscript PDF file)?

Reviewer #1: Yes

4. Is the manuscript presented in an intelligible fashion and written in standard English?

Reviewer #1: Yes

5. Review Comments to the Author

Reviewer #1: 1. Page 4, line 132. “…12-17 years old) stated on July 1, 201, …” . There is a spelling mistake here.

2. Page 5, line 144. “Currently, there is limited research on the decision-making of parents of children under regarding COVID-19 vaccination, particularly in the context of cross-country comparisons “. This sentence is not entirely correct; although there are few studies that make cross-country comparisons, there are many studies on this topic.

3. “Vaccine hesitant” is not defined in the text, but the sources used in the definition are referenced. However, what “vaccine hesitant” means needs to be explained in detail. Otherwise, it is not possible to understand Table 1. For example, in row 1 of Table 1, if the rate of vaccine hesitant is 55%, how can "wants vaccine for self" be 84% and "wants vaccine for child" be 90%?

4. On the right-hand side of Table 1, there are rows whose sum does not add up to 100%: row 5, row 17, row 24, row 28 and row 29.

5. How realistic are the definitions used for the vaccine profile (50% effective with a 20% risk of fever, 50% effective with a 5% risk of fever, etc.)? Is there a reference you use when making these definitions? Is fever the only side effect of the vaccine?

6. The same findings are repeated on pages 11 and 12.

7. What is the reason for including the dates 2021-03 and 2021-06 in the multivariable model in Table 3?

8. Page 13, line 283. “The degree of vaccine preference discordance in children and adults was found to be higher in the United States and Taiwan according to Table 4”. However, In China, the proportion who "want the vaccine for self not child" is 12%, the same as in the US.

9. Table 4 only includes data for June 2021. What is the reason for this?

10. Based on the data in Table 1, a discussion comparing vaccination intention or behavior would be more appropriate. The discussion gives the impression that this study only looked at vaccination intentions. However, the authors have data both before and after countries started vaccination. Looking at the data from this perspective makes the article much more valuable.

11. In the discussion, topics completely different from the aim of the study and the questions investigated (the relationship of other vaccines such as MMR with sociodemographic factors, etc...) are raised. The discussion should focus on the study data.

12. In its present form, the conclusion is irrelevant to the study data and discussion. The conclusion should be based on the results of the study and make a recommendation based on these data.

6. PLOS authors have the option to publish the peer review history of their article (what does this mean?). If published, this will include your full peer review and any attached files.

**Do you want your identity to be public for this peer review?** For information about this choice, including consent withdrawal, please see our Privacy Policy.

Reviewer #1: **Yes: **Gülsüm İclal Bayhan

---

## [Decision Letter · Decision Letter 1]

5 Feb 2024

Parental vaccine hesitancy and acceptance of a COVID-19 vaccine: an internet-based survey in the US and five Asian Countries

PGPH-D-23-01104R1

Dear Dr. Wagner,

We are pleased to inform you that your manuscript 'Parental vaccine hesitancy and acceptance of a COVID-19 vaccine: an internet-based survey in the US and five Asian Countries' has been provisionally accepted for publication in PLOS Global Public Health.

Best regards,

Julia Robinson

Executive Editor

Reviewer Comments (if any, and for reference):

Reviewer's Responses to Questions

**Comments to the Author**

1. If the authors have adequately addressed your comments raised in a previous round of review and you feel that this manuscript is now acceptable for publication, you may indicate that here to bypass the “Comments to the Author” section, enter your conflict of interest statement in the “Confidential to Editor” section, and submit your "Accept" recommendation.

Reviewer #1: All comments have been addressed

Reviewer #2: All comments have been addressed

2. Does this manuscript meet PLOS Global Public Health’s publication criteria? Is the manuscript technically sound, and do the data support the conclusions? The manuscript must describe methodologically and ethically rigorous research with conclusions that are appropriately drawn based on the data presented.

Reviewer #1: Yes

Reviewer #2: Yes

3. Has the statistical analysis been performed appropriately and rigorously?

Reviewer #1: Yes

Reviewer #2: Yes

4. Have the authors made all data underlying the findings in their manuscript fully available (please refer to the Data Availability Statement at the start of the manuscript PDF file)?

Reviewer #1: Yes

Reviewer #2: Yes

5. Is the manuscript presented in an intelligible fashion and written in standard English?

Reviewer #1: Yes

Reviewer #2: Yes

6. Review Comments to the Author

Reviewer #1: (No Response)

Reviewer #2: (No Response)

7. PLOS authors have the option to publish the peer review history of their article (what does this mean?). If published, this will include your full peer review and any attached files.

**Do you want your identity to be public for this peer review?** For information about this choice, including consent withdrawal, please see our Privacy Policy.

Reviewer #1: **Yes: **Gülsüm İclal BAYHAN

Reviewer #2: No
